# Accuracy of point-of-care nasopharyngeal Interleukin 6 and lung ultrasound in predicting the development of bronchopulmonary dysplasia in preterm infants born before 30 weeks of gestation

**Marta Teresa-Palacio**[1☯], **Xela Avià**[2☯], **Carla Balcells-Esponera**[3], **Ana Herranz-Barbero**[1], **Miguel Alsina-Casanova**[1], **Cristina Carrasco**[3], **Maria-Dolors Salvia**[1], **Victoria Aldecoa-Bilbao**[1*]

**1** Neonatology Department, Hospital Clínic Barcelona, BCNatal (Barcelona Center for Maternal Fetal and Neonatal Medicine), University of Barcelona (UB), Barcelona, Spain, **2** Facultat de Ciències de la Salut i de la Vida, Universitat Pompeu Fabra (UPF) de Barcelona, Barcelona, Spain, **3** Neonatology Department, Hospital Sant Joan de Déu, BCNatal (Barcelona Center for Maternal Fetal and Neonatal Medicine), Barcelona, Spain

☯ These authors contributed equally to this work.

* valdecoa@clinic.cat

## Abstract

### Background

Bronchopulmonary dysplasia (BPD) is a common cause of morbidity in preterm infants, leading to long-term respiratory complications and risk of neurodevelopmental impairment. Although it has a multifactorial etiology, local inflammation may play a major role. Objectives. We aimed to analyze the relationship between nasopharyngeal aspirate (NA) interleukin 6 (IL6) levels and clinical and imaging findings of BPD. Methods: Pilot study in preterm infants < 30 weeks. NA was collected at 7 days of life (DOL) and serial lung ultrasounds (LUS) were performed during admission. NA-IL6 levels were measured using an automated electrochemiluminescence immune-analyzer Cobas-e602 and an IL6 ELISA method. Results. Forty-two patients were studied. Infants with BPD had significantly lower gestational age and higher levels of NA-IL6 at DOL 7. Both methods showed good agreement: ICC = 0.937 (95%CI 0.908-0.957); p < 0.001) and Passing-Bablok Regression ($R^2$ = 0.961). LUS score (AUC = 0.83) and NA-IL6 (AUC = 0.81) at DOL 7 predicted BPD. The AUC of NA-IL6 as a stand-alone marker of BPD was 0.808 (95% CI 0.67 – 0.94); p = 0.002, with 24 pg/ml being the best cutoff with a sensitivity and specificity of 83.3%. A model including birth weight, LUS score at DOL7, NA-IL6 at DOL7, and days of mechanical ventilation predicted BPD with $R^2$ = 0.600 (p < 0.001). Conclusions. Point-of-care assessment of NA-IL6 is feasible and reliable compared with a reference method and can be useful in managing BPD. Predictive models of BPD in the first week of life, including clinical, biological, and imaging biomarkers must be tested in larger cohorts.

**Data availability statement:** All data used in this study is available at https://doi.org/10.6084/m9.figshare.28514213.v1.

**Funding:** This study was supported by the Spanish Neonatal Society (2020 "RespiSurf Adolfo Valls" grant and in 2022 Investigation grant). The funders had no role in study design, data collection and analysis, decision to publish, or preparation of the manuscript. There was no additional external funding received for this study.

**Competing interests:** The authors have declared that no competing interests exist.

## Introduction

Bronchopulmonary dysplasia (BPD) is a clinical lung injury syndrome that disrupts alveolarization and microvascular development. It remains a significant source of morbidity in extremely preterm infants despite remarkable advancements in neonatal care in recent decades [1,2]. Its prevalence is still considerable, leading to pulmonary complications and imposing a substantial economic burden on both families and society [3,4]. The pathogenesis of BPD is complex and involves prenatal and postnatal insults to the developing lung leading to different phenotypes [5]. According to the current definitions, diagnosis is established at 36 weeks postmenstrual age (PMA), which limits the implementation of preventive measures [6–8]. Hence, discovering reliable clinical, biological, and imaging markers predictive of BPD in preterm infants would enhance diagnostic and therapeutic strategies.

Studies performed in large cohorts have built predictive models for BPD using only clinical variables that are readily available immediately after birth. Although most of these studies showed good accuracy in predicting BPD, only a few have validated their models [9–11]. Moreover, owing to the retrospective nature of the data, the refinement of BPD definitions, and the improvement in the care of extremely preterm infants over time, their applicability in clinical practice seems to be diminished.

Local inflammation plays a pivotal role in BPD [2,12]; however, the precise mechanisms remain incompletely understood, hampering the development of effective preventive and therapeutic strategies [1,7,13]. Among inflammatory mediators, interleukin 6 (IL6) plays a significant role in this process instigating inflammation in premature lungs, and has been proposed as a potential biomarker for BPD [14,15]. Previous clinical investigations have revealed elevated levels of several cytokines in the blood, tracheal aspirates, and bronchoalveolar lavage fluid of infants at risk of developing BPD [13,14,16–19]. However, the invasive nature of sample collection methods poses potential risks to vulnerable patients. Moreover, obtaining samples such as tracheal aspirates or bronchoalveolar lavage has become increasingly challenging owing to the declining proportion of intubated preterm infants. Thus, researchers are exploring non-invasive techniques for cytokine detection, such as cord blood, salivary samples, and nasopharyngeal aspirates (NA) [20–22].

Most studies have utilized the enzyme-linked immunosorbent assay (ELISA) method to measure IL6 concentrations, which, although highly sensitive, involves a manual and labor-intensive process, leading to delayed results. Bedside tests have been developed to provide immediate results, enabling rapid clinical decision-making and more tailored treatments and interventions [23,24]. IL6 point-of-care (POC) tests, including automated electrochemiluminescence immunoassays such as Cobas®, have demonstrated their utility in different clinical settings [23,25,26].

Diagnostic imaging is essential in BPD, not only for clinical management but also for establishing a more accurate prognosis in the long term. Chest X-ray remains the most used imaging test in Neonatal Intensive Care Units (NICU), but it is a poor predictor of progression to BPD in most cases. In recent years, techniques such as computed tomography and, more recently, thoracic resonance imaging have proven to be useful in evaluating the immature lung and distinguishing between the different BPD phenotypes [27–30], but they are not widely available. Conversely, lung ultrasound (LUS) is a non-invasive and useful technique that is increasingly used in the NICU to diagnose and manage neonatal respiratory diseases [31]. Recent studies have also suggested that LUS has potential as a promising tool for BPD [32–35].

In this preliminary study, we aimed to prove the feasibility of a POC method to assess IL6 levels in NA among preterm infants, analyze its correlation with LUS findings, and evaluate its effectiveness in predicting BPD.

## Materials and methods

### Study design and patient population

This longitudinal prospective study included all preterm infants consecutively admitted to the Neonatology Department of the Hospital Clínic of Barcelona between 01/12/2020 and 30/06/2022. Inborn patients with a gestational age between $23^0$ and $29^6$ weeks were eligible for recruitment. Exclusion criteria were major congenital malformations, or unavailability of the investigators to perform the ultrasounds or to process the NA.

Patient selection and sampling procedures were performed following the Declaration of Helsinki and applicable local regulatory requirements after approval from the Institutional Review Boards (reference number HCB/2020/1395). Before enrollment, written informed consent was obtained from parents or guardians.

### Respiratory management

Delivery room stabilization included delay cord clamping and plastic wrap. Spontaneously breathing infants were stabilized with humidified CPAP via face mask. Intermittent positive pressure (peak inspiratory pressure of 20–25 $cmH_2O$ and positive end-expiratory pressure of 5 $cmH_2O$) was started in apneic or bradycardic infants. Initial $FiO_2$ was 30% and then titrated to achieve oxygen saturation of 90% at 10 min of life based on preductal pulse oximetry. On admission, CPAP was continued via nasal mask with a positive end-expiratory pressure between 6 and 8 $cmH_2O$. The oxygen saturation target range was 90% - 95%. Intravenous caffeine and parenteral nutrition were started at admission. Surfactant therapy (200 mg/kg; Curosurf®, Chiesi Pharmaceuticals) was administered according to the European guidelines [36], using the less invasive surfactant administration method in infants with CPAP. Echocardiographic screening for patent ductus arteriosus (PDA) was requested at the discretion of the neonatologist in charge of the patient. Respiratory management was made according to local protocol and postnatal steroids were considered in case of more than 7 consecutive days of mechanical ventilation (MV) beyond the second week of life.

### Sample collection and IL6 analysis

NA was collected around the 7th day of life (DOL) (between DOL 7 and 10) and processed using a standardized technique by trained hospital personnel [37]. A soft nasal catheter (silicone, 5 Fr size) was introduced through one nostril into the nasopharynx and was suctioned into 1mL of sterile saline solution. This procedure takes approximately less than 10 seconds. Samples were centrifuged, and the supernatant was kept at -80°C until its analysis after the recruitment of all patients. IL6 levels in NA, measured in pg/ml, were determined using two different methods at the same time: an automated Cobas e602 electrochemiluminescence immunoanalyzer (Cobas 8000 platform®, Roche Diagnostics, Switzerland) and an ELISA method (DiaSource®, Louvain-la-Neuve, Belgium).

### Lung ultrasound evaluations

Bedside LUS were performed at admission (between 60 and 120 minutes of life), at 3, 7, 28 DOL, and 36 weeks of PMA. A Sonosite® X-Porte machine and a linear probe (VF 13-5MHz) were used. Patients were in a supine position. Each lung was divided into three areas of the thorax defined by the midclavicular (anterior), the anterior axillary line (lateral), and the posterior axillary line (posterior) and through longitudinal orientation. The complete LUS protocol has previously been described elsewhere [33,38]. LUS scores (0-18) in each area as described by Brat et al. [39] were calculated by two neonatologists with LUS experience at the end of the recruitment period.

## Variables and outcomes definitions

Demographic data, respiratory variables, and main outcomes during admission were collected. MV, days of oxygen, oxygen saturation, $FiO_2$, type of respiratory support, and mean airway pressure were recorded at each time point. The time of surfactant administration, the need of postnatal steroids, and the duration of respiratory support at discharge were also documented. BPD was defined according to the mode of respiratory support at 36 weeks of PMA, regardless of supplemental oxygen use as published by Jensen et al. [8]. Intraventricular hemorrhage was graded according to Papile's classification [40]; necrotizing enterocolitis (NEC) was defined using Bell's classification [41] and retinopathy of prematurity (ROP) was classified according to the International Committee for Classification of ROP [42].

## Sample size and statistical analysis

The sample size calculation was made based on the expected accuracy of the diagnostic method (nasopharyngeal interleukin 6) to predict BPD using the area under the curve receiver-operating characteristic (AUC). To obtain an AUC = 0.83 with a marginal error (d) of 0.10 and a 95% confidence level, we estimate a sample size of 42 subjects [43].

We described the variables measured as mean and standard deviation if normally distributed or as the median and interquartile range (25th-75th centile) if not. Univariate analysis (Chi-square test or Fisher's exact test) was performed for categorical comparisons, and *t*-Student or Mann-Whitney test for continuous variables. IL6 values were log-transformed. The Spearman or Pearson test as appropriate was used to assess the correlation between IL6 and LUS scores at DOL 7.

The area under the curve receiver-operating characteristic (AUC) with its 95% confidence interval (CI) and sensitivity, specificity, positive predictive value (PPV), negative predictive value (NPV), positive likelihood ratio (LR+), and negative likelihood ratio (LR−) were calculated for each predictor. The intraclass correlation coefficient (ICC) with its 95% CI and Passing-Bablok regression analyses were performed to assess the agreement between both methods [44].

Bilateral p-values inferior to 0.05 were accepted as statistically significant. Statistical analysis was performed using SPSS 22.0 (IBM Corporation, USA) and the Graph Pad Prism v.5 program for the graphs.

## Results

Over the study period, 67 preterm infants born before 30 weeks of gestation were eligible for recruitment, and 42 were included (Fig 1). No complications were observed from catheter insertion for NPA collection. Their clinical baseline characteristics are summarized in Table 1.

Levels of NA-IL6 were also significantly higher in those with BPD, 40.0 [17.5 - 186] vs 4.0 [2 – 13]; p < 0.05. Levels of NA-IL6 and LUS scores at DOL 7 were not correlated (*Spearman's rho* = 0.249; p = 0.112). Preterm infants with BPD significantly needed more days of MV and postnatal steroids and exhibited poorer outcomes (Table 2).

Patients treated with postnatal steroids exhibited higher LUS scores at DOL 7 with a median of 6 [4 – 8] vs 12 [11.5 – 14.5]; p = 0.016. Levels of NA-IL6 in those patients were slightly higher but not significant, with a median of 38.0 [2 – 3932] vs 7.0 [1 – 74]; p = 1.00.

Both laboratory methods showed a good agreement with an ICC = 0.937 (95% CI 0.908 – 0.957); p < 0.001, and with a $R^2$ = 0.961 using Passing-Bablok regression analysis (Fig 3).

The predictive model that combines the best variables for predicting BPD is detailed in Table 3.

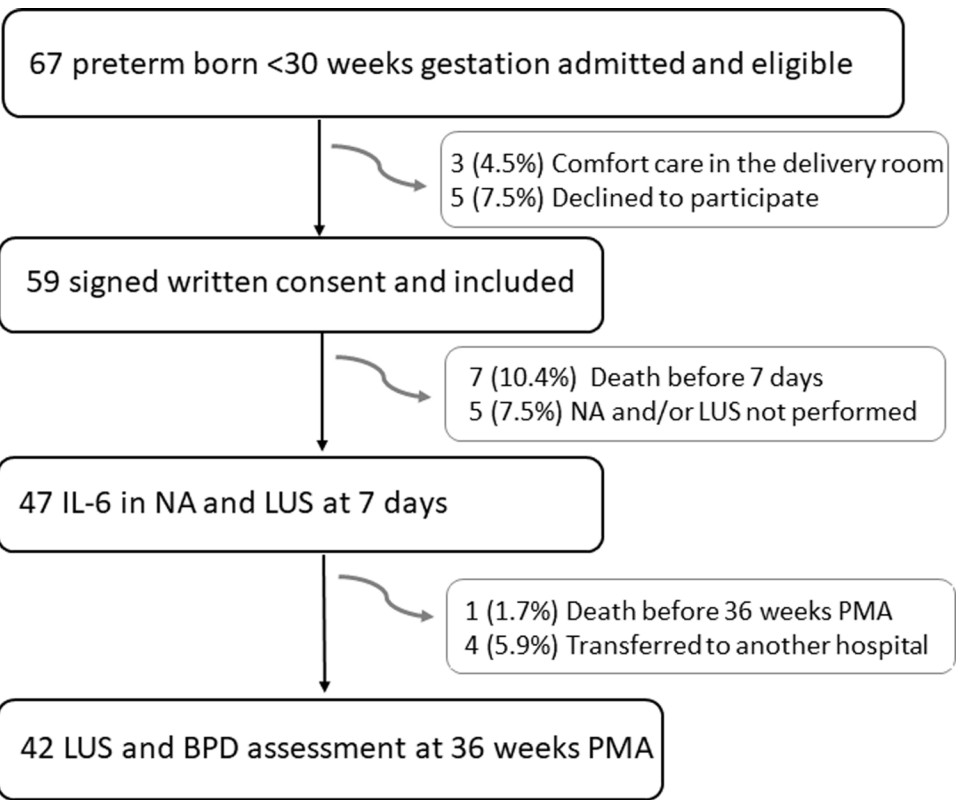

**Fig 1. Flowchart of the study population.**

The AUC of NA-IL6 as a stand-alone marker of BPD was 0.808 (95% CI 0.67 – 0.94); p = 0.002, with 24 pg/ml being the best cutoff with a sensitivity and specificity of 83.3%, false-positive rate of 26.7%, PPV of 66.7%, NPV of 92.6%, LR+ of 5.00, and LR− of 0.20.

## Discussion

This pilot study provides valuable information regarding a non-invasive method to detect IL6 levels in NA in preterm neonates. We also demonstrate a good correlation with LUS findings as potential predictors of BPD. Moreover, we conducted a comparison between the IL6 levels in NA obtained through ELISA and those obtained through an automated immunoassay, revealing a strong correlation between them as we illustrate in Fig 3. Although this correlation has been previously observed in various settings involving plasmatic IL6 levels in sepsis [23,24,45] and in intraamniotic fluid [26], we found that ELISA and Cobas were also comparable for measuring IL6 in a readily available respiratory matrix in preterm infants at risk of developing BPD. Nevertheless, from a clinical standpoint, Cobas exhibits greater advantages.

In the past, many studies have investigated the inflammatory profile in patients evolving to BPD by assessing cord blood [22,46,47], respiratory secretions using bronchoalveolar lavage [14,15,19,48,49], and saliva [21]. Although most of these studies demonstrated different concentrations of cytokines and other mediators between preterm infants with and without BPD, their heterogeneity in methodology and time-point assessments, respiratory management, and BPD definitions hinder their translation to the clinical practice. More recently, studies in the pediatric population have explored the cytokines profile in NA, in the context of viral or bacterial respiratory infectious diseases [20,50] but, to our knowledge, NA-IL6 has not

**Table 1. Baseline characteristics and respiratory status in the first week of life in preterm infants born less than 30 weeks gestation according to the diagnosis of bronchopulmonary dysplasia.**

|  | Bronchopulmonary dysplasia | | |
|---|---|---|---|
|  | No (n = 30) | Yes (n = 12) | p-value |
| Gestational age (weeks) | 27.8 [26.8-28.9] | 27.2 [25.8-27.6] | 0.088 |
| Birth weight (grams) | 993 [840-1054] | 690 [575-913] | 0.017* |
| Male sex | 13 (43.3) | 6 (50.0) | 0.695 |
| Antenatal steroids | 27 (90.0) | 11 (91.7) | 0.868 |
| Cesarean delivery | 17 (56.7) | 8 (66.7) | 0.551 |
| IUGR | 8 (26.7) | 6 (50.0) | 0.147 |
| Olygoamnios | 7 (23.3) | 1 (8.3) | 0.263 |
| Apgar score (5 minutes) | 9 [8–10] | 8 [7–9] | 0.372 |
| CRIB-II score | 9 [7–10] | 12 [9–14] | 0.066 |
| Surfactant administration | 12 (40.0) | 6 (50.0) | 0.554 |
| Surfactant (hours of life) | 3 [0.6-3.5] | 2 [1.5-3.0] | 0.620 |
| Early onset sepsis | 2 (6.7) | 1 (8.3) | 0.850 |
| LUS score (60-120 min) | 6 [4 – 10] | 8 [6 – 10.8] | 0.462 |
| LUS score (3 days) | 8 [6–10] | 8 [6–12] | 0.900 |
| LUS score (7 days) | 5.5 [4–8] | 11 [6.5-12] | 0.011* |
| IL-6 (pg/ml) in NA (7 days) | 4.0 [2–13] | 40 [18-186] | 0.017* |
| PDA | 14 (46.7) | 6 (50.0) | 0.845 |

Values are expressed as median [25th-75th centile] or number (%). *p-value < 0.05.

Abbreviations. IL: Interleukin. IUGR: intrauterine growth restriction; LUS: lung ultrasound; PDA: patent ductus arteriosus; NA: nasopharyngeal aspirate.

Twelve infants developed BPD (28.6%). When comparing both cohorts, we found higher LUS scores at 7 and 28 DOL, but not on days 1 or 3 (Fig 2).

been studied in the context of BPD. These studies have proved that NA is a reliable biological matrix for assessing airway and lung inflammation. Among other pro-inflammatory mediators, IL 6 plays a significant role in instigating inflammation in premature lungs and has been proposed as a potential biomarker for BPD. It acts by suppressing neutrophil migration and activating mononuclear cells [15] and may also have direct detrimental effects on structures of lung tissue by affecting cell integrity and inducing apoptosis [14].

Our results showed that infants with higher IL6 levels in NA on DOL 7 were at a higher risk of developing BPD with an AUC of 0.81 and a cut-off point of 24 pg/ml. Despite this promising finding, IL6 levels alone may not be a sensitive or specific marker to predict the development of BPD if not combined with other variables. However, NA-IL6 evaluation in the context of BPD with a rapid and reliable method, can be useful for monitoring an evolving BPD and for treatment assessment. Preterm infants exhibiting higher NA-IL6 levels and higher LUS scores in the first weeks of life could benefit the most from anti-inflammatory treatments such as systemic steroids [51]. Moreover, these two markers may be useful for assessing the local steroid response when administered with surfactant.

Another key finding of this study, consistent with previous publications, is that infants with a higher LUS score at DOL 7 and DOL 28 were more likely to develop BPD. Several studies and a meta-analysis have assessed the accuracy of LUS in predicting BPD, showing an AUC greater than 0.85 [33,52]. The accuracy of LUS score at DOL 7 found in our study (AUC = 0.83) is slightly inferior and may be explained by the fact that in our previous study, LUS evaluations were blinded for clinicians. POC ultrasound was successfully implemented in

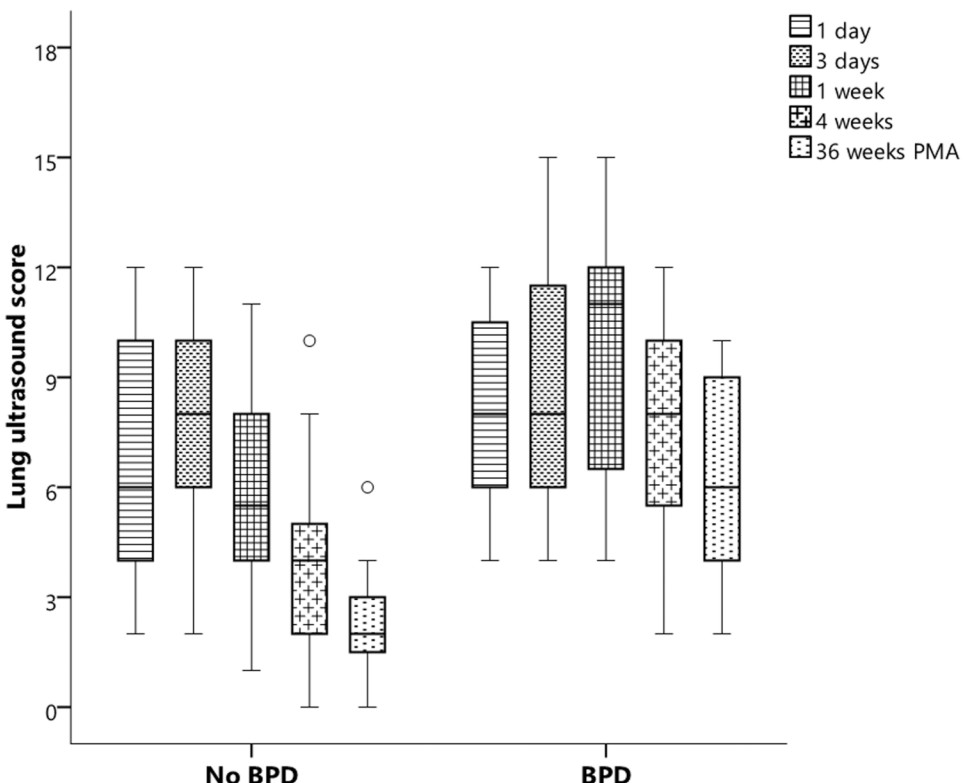

**Fig 2. Median lung ultrasound scores and [25th – 75th centiles] during admission according to bronchopulmonary dysplasia diagnosis in preterm infants born at < 30 weeks of gestation.**

**Table 2. Respiratory and main outcomes at discharge according to the diagnosis of bronchopulmonary dysplasia.**

|  | Bronchopulmonary dysplasia | | |
| --- | --- | --- | --- |
|  | No (n = 30) | Yes (n = 12) | p-value |
| Mechanical ventilation | 7 (23.3) | 7 (58.3) | 0.030* |
| Mechanical ventilation > 5 days | 1 (3.3) | 5 (41.7) | 0.001* |
| CPAP (days) | 5.5 [2-10.5] | 27 [5.5-83] | 0.088 |
| Oxygen therapy (days) | 9 [1-50] | 88 [66-106] | <0.001* |
| Late-onset sepsis | 2 (6.7) | 2 (16.7) | 0.319 |
| Antibiotic exposure (days) | 2.0 [0-3] | 3.5 [2-11.5] | 0.175 |
| Blood transfusion | 9 (30.0) | 9 (75.0) | 0.008* |
| Postnatal steroids | 1 (3.3) | 4 (33.3) | 0.007* |
| Surgical NEC or SIP | 0 (0.0) | 2 (16.7) | 0.022* |
| ROP | 9 (30.0) | 8 (66.7) | 0.029* |
| IVH | 5 (16.7) | 2 (16.7) | 1.000 |
| LUS score at 28 days | 4 [2–5] | 8 [4.3-10] | 0.024* |
| LUS score at 36 weeks PMA | 2 [1–4] | 6 [3–10] | 0.007* |
| Length of stay (days) | 56 [46-68] | 95 [74-111] | 0.002* |

Values are expressed as median [25th-75th centile] or number (%). *p-value < 0.05.

Abbreviations. CPAP: continuous positive airway pressure; IVH: intraventricular hemorrhage; LUS: lung ultrasound; NEC: necrotizing enterocolitis; PMA: postmenstrual age; ROP: retinopathy of prematurity; SIP: spontaneous intestinal perforation.

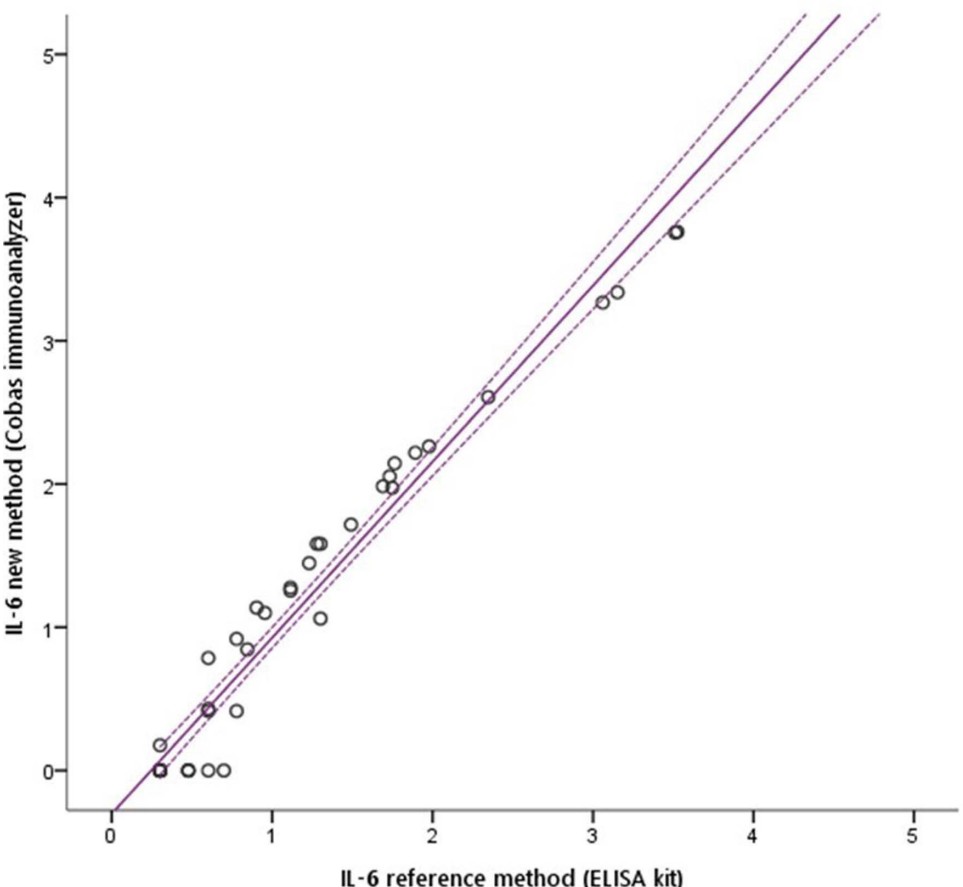

**Fig 3. Comparison of the two methods (Point-of-care Cobas® vs ELISA reference method) using Passing-Bablok regression analysis to assess interleukin 6 in nasopharyngeal aspirates at 7 days of life in preterm infants born at <30 weeks gestation.**

**Table 3. Predictors of bronchopulmonary dysplasia in preterm infants born less than 30 weeks gestation.**

| Variables | AUC | 95% CI | p-value |
|---|---|---|---|
| LUS score DOL 7 | 0.829 | 0.69 – 0.97 | 0.001 |
| NA IL-6 DOL 7 | 0.808 | 0.67 – 0.94 | 0.002 |
| Birth weight (grams) | 0.799 | 0.63 – 0.97 | 0.003 |
| Days of mechanical ventilation | 0.725 | 0.53 – 0.92 | 0.024 |

$R^2$ = 0.600; p-value <0.001 according to the likelihood ratio test.

Abbreviations. AUC: Area Under the Curve; CI: confidence interval; DOL: day of life; IL: Interleukin; LUS: lung ultrasound; NA: nasopharyngeal aspirate; OR: odds ratio; $R^2$: Nagelkerke's coefficient.

our NICU in 2018, and LUS is performed almost every day to guide respiratory management in ventilated preterm infants. LUS findings, and not only work of breathing or oxygen needs, are considered to guide lung recruitment, prone positioning, and respiratory physiotherapy in selected patients to avoid atelectasis. This change in respiratory care and other BPD "bundles", may be traduced in less chronic lung disease in preterm infants in the future. Our results showed that the combination of two simple variables, LUS score and NA-IL6, both easily

available at the bedside in the first week of life, explains 60% of the variance of the main outcome of the study. These two variables may identify those patients with a low risk of BPD (low NA-IL6 and low LUS score), and those with evolving BPD (high NA-IL6 and high LUS score) from those with a different BPD pattern. Differentiating groups of high-risk of moderate to severe BPD will allow us to treat them in a tailored manner [53]. A multivariate predictive model including these variables in combination with the clinical variables that have shown good accuracy in predicting BPD in previous studies deserves further investigation [10,54].

BPD is a multifactorial disease with different phenotypes that require individualized and interdisciplinary approaches and innovative predictive methods [55,56]. Most of the studies published to date have used the BPD definition as a surrogate of the respiratory burden of prematurity [8]. Improving preterm respiratory health in the long term probably requires a more accurate endpoint when assessing different interventions.

We acknowledge some limitations of this study due to its small sample size. Future studies with larger data should be performed to validate these findings. Furthermore, we do not know the impact of intraamniotic inflammation and/or infection on the values of IL6 in NA in the first week of life, nor the impact of healthcare-related pneumonia that can occur typically after 7-10 days of MV in extremely preterm infants. Moreover, we did not consider other definitions of BPD as well as the possible modulator effect of postnatal steroids on the main outcome of the study, the need for respiratory support at 36 weeks PMA.

## Conclusion

In this pilot study, POC assessment of NA-IL6 has shown to be feasible in preterm infants and reliable compared with a reference method, and it can be useful in managing BPD. The early prediction of BPD in high-risk patients with rapid and non-invasive biomarkers provides a window of opportunity for tailored treatments. To prevent BPD development, multicentric prospective studies in larger cohorts are needed to develop POC predictive models integrating clinical, biological, and echographic findings.

## Author contributions

**Conceptualization:** Marta Teresa-Palacio, Xela Avià, Maria-Dolors Salvia, Victoria Aldecoa-Bilbao.

**Data curation:** Marta Teresa-Palacio, Xela Avià, Carla Balcells-Esponera, Ana Herranz-Barbero, Miguel Alsina-Casanova, Cristina Carrasco, Victoria Aldecoa-Bilbao.

**Formal analysis:** Marta Teresa-Palacio, Xela Avià, Victoria Aldecoa-Bilbao.

**Funding acquisition:** Maria-Dolors Salvia, Victoria Aldecoa-Bilbao.

**Investigation:** Marta Teresa-Palacio, Xela Avià, Carla Balcells-Esponera, Ana Herranz-Barbero, Miguel Alsina-Casanova, Cristina Carrasco, Maria-Dolors Salvia, Victoria Aldecoa-Bilbao.

**Methodology:** Marta Teresa-Palacio, Xela Avià, Carla Balcells-Esponera, Ana Herranz-Barbero, Maria-Dolors Salvia, Victoria Aldecoa-Bilbao.

**Project administration:** Marta Teresa-Palacio, Xela Avià, Carla Balcells-Esponera, Ana Herranz-Barbero, Miguel Alsina-Casanova, Maria-Dolors Salvia, Victoria Aldecoa-Bilbao.

**Supervision:** Carla Balcells-Esponera, Ana Herranz-Barbero, Miguel Alsina-Casanova, Cristina Carrasco, Maria-Dolors Salvia, Victoria Aldecoa-Bilbao.

**Validation:** Cristina Carrasco, Maria-Dolors Salvia, Victoria Aldecoa-Bilbao.

**Writing – original draft:** Marta Teresa-Palacio, Xela Avià.

**Writing – review & editing:** Marta Teresa-Palacio, Xela Avià, Carla Balcells-Esponera, Ana Herranz-Barbero, Miguel Alsina-Casanova, Cristina Carrasco, Maria-Dolors Salvia, Victoria Aldecoa-Bilbao.

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
