## [Decision Letter · Decision Letter 0]

1 Nov 2024

PONE-D-24-31846Accuracy of point-of-care nasopharyngeal Interleukin 6 and lung ultrasound in predicting the development of bronchopulmonary dysplasia.PLOS ONE

Dear Dr. Aldecoa-Bilbao,

Thank you for submitting your manuscript to PLOS ONE. After careful consideration, we feel that it has merit but does not fully meet PLOS ONE’s publication criteria as it currently stands. Therefore, we invite you to submit a revised version of the manuscript that addresses the points raised during the review process.

Overall, the reviewers supported your paper for publication. However, several critiques are raised from reviewers to be resolved. Please revised the paper according to their advice.

We look forward to receiving your revised manuscript.

Kind regards,

Kazumichi Fujioka

Academic Editor

PLOS ONE

Journal requirements: When submitting your revision, we need you to address these additional requirements. 1. Please ensure that your manuscript meets PLOS ONE's style requirements, including those for file naming. The PLOS ONE style templates can be found at https://journals.plos.org/plosone/s/file?id=wjVg/PLOSOne_formatting_sample_main_body.pdf and https://journals.plos.org/plosone/s/file?id=ba62/PLOSOne_formatting_sample_title_authors_affiliations.pdf 2. We note that you have indicated that there are restrictions to data sharing for this study. For studies involving human research participant data or other sensitive data, we encourage authors to share de-identified or anonymized data. However, when data cannot be publicly shared for ethical reasons, we allow authors to make their data sets available upon request. For information on unacceptable data access restrictions, please see http://journals.plos.org/plosone/s/data-availability#loc-unacceptable-data-access-restrictions.  Before we proceed with your manuscript, please address the following prompts: a) If there are ethical or legal restrictions on sharing a de-identified data set, please explain them in detail (e.g., data contain potentially identifying or sensitive patient information, data are owned by a third-party organization, etc.) and who has imposed them (e.g., a Research Ethics Committee or Institutional Review Board, etc.). Please also provide contact information for a data access committee, ethics committee, or other institutional body to which data requests may be sent. b) If there are no restrictions, please upload the minimal anonymized data set necessary to replicate your study findings to a stable, public repository and provide us with the relevant URLs, DOIs, or accession numbers. Please see http://www.bmj.com/content/340/bmj.c181.long for guidelines on how to de-identify and prepare clinical data for publication. For a list of recommended repositories, please see https://journals.plos.org/plosone/s/recommended-repositories. You also have the option of uploading the data as Supporting Information files, but we would recommend depositing data directly to a data repository if possible. Please update your Data Availability statement in the submission form accordingly.

Reviewers' comments:

Reviewer's Responses to Questions

**Comments to the Author**

1. Is the manuscript technically sound, and do the data support the conclusions?

Reviewer #1: Yes

Reviewer #2: Yes

Reviewer #3: Partly

Reviewer #4: Partly

2. Has the statistical analysis been performed appropriately and rigorously? 

Reviewer #1: Yes

Reviewer #2: Yes

Reviewer #3: Yes

Reviewer #4: I Don't Know

3. Have the authors made all data underlying the findings in their manuscript fully available?

Reviewer #1: Yes

Reviewer #2: Yes

Reviewer #3: Yes

Reviewer #4: No

4. Is the manuscript presented in an intelligible fashion and written in standard English?

Reviewer #1: Yes

Reviewer #2: Yes

Reviewer #3: Yes

Reviewer #4: Yes

5. Review Comments to the Author

Reviewer #1: I read with great interest the study titled “Accuracy of point-of-care nasopharyngeal Interleukin 6 and lung ultrasound in predicting the development of bronchopulmonary dysplasia” where the authors studied the feasibility of a POC assessment of IL6 levels in nasopharyngeal sample then examined its correlation with LUS findings as early predictors of BPD among preterm infants. Below are my key points of feedback:

Title: please add “in preterm infants (or in infants < 30 weeks’ GA)”

Abstract: No concerns

Introduction: well-written but long. I suggest you reduce it by one third. For example, you can delete from line 80- 87.

Methods: well-written

Results: well-written, no concerns

Conclusion: I strongly recommend you start with; “In this pilot study, ……..” OR “In this small cohort of preterm infants,…………”

Reference: Please use most recent reference for ROP classification: DOI: 10.1016/j.ophtha.2021.05.031

Reviewer #2: Dear Authors

this is a very interesting paper with an innovative idea to try to predict BPD.

The main disadvantage of the study is the very small size of BPD population. I think this point need to be more exstensively andressed in Discussion. Did to try to calculate a sample size needed for the study?

Also I think need to be more exstensively andressed in Discussion the interpretation of the result of the R2 of the prediction model (R2 = 0.6)

You should compare the power of prediction of your model with other BPD prediction model published in litterature in Discussion

Reviewer #3: This is a novel paper that uses a combination of nasal IL-6, which is relatively minimally invasive, and LUS findings to predict BPD. IL-6 measurement using Cobas, a fully automated analyzer, is simple and showed a correlation with ELISA. The AUC values for LUS and NA-IL6 are high, and I think that the cutoff value for NA-IL6 of 24 pg/ml for BPD could be a clinically valuable indicator.

As the authors also noted in the Limitations section, I think the sample size is small. It is important to increase the number of cases and use multivariate analysis to investigate the relationship between BPD and covariates such as IL-6, LUS score, gestational age, CAM, and oxygen administration.

Method

Please define BPD and provide references.

Discussion

Overall, there is not enough discussion.

Please describe the mechanism by which cytokines including IL-6 are involved in BPD. Are there any previous reports on the association between nasal IL-6 and BPD? If there are no previous reports examining the association between nasal IL-6 and BPD, it would further enhance the value of this study.

Please describe the safety of nasal discharge collection in very low birth weight infants. Because the nasal cavity is narrow, it would be helpful to describe whether there were any complications when inserting the catheter into the nasopharynx (e.g., nasal cavity damage, nosebleeds, desaturation, etc.). Also, please mention the product name of the catheter used.

Reviewer #4: This is a clinical study conducted to provide lung ultrasound and nasopharyngeal interleukin-6 would be predicting marker of the development of BPD.

The topic is important and the manuscript is well written, but there are some concerns.

Major points

1. Concerns about very limited sample size. Further, the author should perform power analysis.

2. Chorioamnionitis is known to be associated with BPD. The author has to clarify the relationship between chorioamnionitis and NA IL-6 in this study. This leads whether NA IL-6 elevation is due to prenatal or postnatal factors.

3. Multivariate analysis should be needed to clarify whether LUS and IL-6 associate with BPD.

4. No description about Figure 3 was found in discussion.

Minor points

1. Figure 2: There are no notations of the median or mean.

2. Figure 3: How many samples did the author investigated NA-IL6? There are almost 35 dots in this Figure.

3. SGA was associated with risk of BPD in this study?

6. PLOS authors have the option to publish the peer review history of their article (what does this mean? ). If published, this will include your full peer review and any attached files.

**Do you want your identity to be public for this peer review?** For information about this choice, including consent withdrawal, please see our Privacy Policy .

Reviewer #1: **Yes: ** Adel Mohamed

Reviewer #2: No

Reviewer #3: No

Reviewer #4: No

---

## [Author Response · Author response to Decision Letter 1]

23 Dec 2024

Dear Editor and reviewers,

Thank you for considering our manuscript and for your kind invitation to submit a revision. I am sorry it took me so long to respond, longer than I would have liked, but it is due to special circumstances as the first author of this manuscript has been on maternity leave since we submitted the first manuscript.

We appreciate the valuable feedback provided by the editor and the reviewers. According to them, the manuscript has been significantly revised and, hopefully, improved.

Please find below our point-by-point response to each comment (in blue font), including an explanation of all changes made to the original article.

Reviewer #1

I read with great interest the study titled “Accuracy of point-of-care nasopharyngeal Interleukin 6 and lung ultrasound in predicting the development of bronchopulmonary dysplasia” where the authors studied the feasibility of a POC assessment of IL6 levels in nasopharyngeal sample then examined its correlation with LUS findings as early predictors of BPD among preterm infants. Below are my key points of feedback:

Title: please add “in preterm infants (or in infants < 30 weeks’ GA)”

Abstract: No concerns

Introduction: well-written but long. I suggest you reduce it by one third. For example, you can delete from line 80- 87.

Methods: well-written

Results: well-written, no concerns

Conclusion: I strongly recommend you start with; “In this pilot study, ……..” OR “In this small cohort of preterm infants,…………”

Reference: Please use most recent reference for ROP classification: DOI: 10.1016/j.ophtha.2021.05.031

Thank you for the comments. We have changed the running title, introduction, and conclusion as per Reviewer #1’s suggestions. We have also updated the reference for ROP classification for the most recent one (reference number 42).

Reviewer #2:

Dear Authors

this is a very interesting paper with an innovative idea to try to predict BPD.

The main disadvantage of the study is the very small size of BPD population. I think this point need to be more exstensively andressed in Discussion. Did to try to calculate a sample size needed for the study?

Thanks for the comment. We are aware that the small size of the sample is an important limitation of the study as we have already mentioned in the manuscript. A pilot study is defined as “the first step of the entire research protocol and is often a smaller-sized study assisting in planning and modification of the main study” [*]. We designed this pilot study to evaluate the feasibility and effectiveness of IL6 levels in predicting BPD before performing larger studies.

Despite being a pilot study, the sample size calculation was made based on the expected accuracy of the diagnostic method (nasopharyngeal interleukin 6) to predict BPD using the area under the curve receiver-operating characteristic (AUC). To obtain an AUC=0.83 with a marginal error (d) of 0.10 and a 95% confidence level, we estimate a sample size of 42 subjects [**] (according to Table 5 estimates in the article Hajian-Tilaki K. Sample size estimation in diagnostic test studies of biomedical informatics. J Biomed Inform. 2014 Apr;48:193-204. doi: 10.1016/j.jbi.2014.02.013).

With the promising results obtained in this work, we will be able to conduct a study with a larger sample size which allows us to validate them. Furthermore, we can perform a multivariate analysis to investigate the relationship between BPD and other covariates as Reviewer #3 also suggests.

We have emphasized this information in the manuscript:

- Abstract section, page 2 , line 30 .

- Introduction section, page 4, line .92

- Methods section, page 7, line 151

- Discussion, page 11, line 215.

References

[*] In J. Introduction of a pilot study. Korean J Anesthesiol. 2017 Dec;70(6):601-605. doi: 10.4097/kjae.2017.70.6.601.

[**] Hajian-Tilaki K. Sample size estimation in diagnostic test studies of biomedical informatics. J Biomed Inform. 2014 Apr;48:193-204. doi: 10.1016/j.jbi.2014.02.013.

Also I think need to be more exstensively andressed in Discussion the interpretation of the result of the R2 of the prediction model (R2 = 0.6)

Thanks for the useful suggestion. We have extended this information in the manuscript. Please kindly see the added text in the Discussion section, page 13, lines 268-276.

Our results showed that the combination of two simple variables, LUS score and NA-IL6, both easily available at the bedside in the first week of life, explains 60% of the variance of the main outcome of the study. These two variables may identify those patients with a low risk of BPD (low NA-IL6 and LUS score), and those with evolving BPD (high NA-IL6 and high LUS score) from those with a different BPD pattern. Differentiating groups of high-risk of moderate to severe BPD will allow us to treat them in a tailored manner[53]. A multivariate predictive model including these two variables with the clinical variables that have shown good accuracy in predicting BPD in previous studies deserves further investigation.

You should compare the power of prediction of your model with other BPD prediction model published in literature in Discussion.

Thank you for your thoughtful suggestion.

BPD prediction models published in the past years are based on perinatal and clinical variables alone or in combination with imaging or biological variables. Despite the heterogeneity of these studies (case mix population and different respiratory management), only a few of them found good AUC values (above 0.80) like those found in our study.

However, none of them has proven yet to be useful in decreasing BPD rates. The goal of our study was to prove that a simple non-invasive point-of-care biomarker can identify preterm infants with sustained inflammation beyond the first week of life. Shortly we will be able to build a predictive model combining clinical variables, lung ultrasound findings, and nasopharyngeal IL6 values at 7 days of life to identify those infants at higher risk of moderate to severe BPD.

We have added these comments in the manuscript (page 14, lines 280-283).

Reviewer #3:

This is a novel paper that uses a combination of nasal IL-6, which is relatively minimally invasive, and LUS findings to predict BPD. IL-6 measurement using Cobas, a fully automated analyzer, is simple and showed a correlation with ELISA. The AUC values for LUS and NA-IL6 are high, and I think that the cutoff value for NA-IL6 of 24 pg/ml for BPD could be a clinically valuable indicator.

As the authors also noted in the Limitations section, I think the sample size is small. It is important to increase the number of cases and use multivariate analysis to investigate the relationship between BPD and covariates such as IL-6, LUS score, gestational age, CAM, and oxygen administration.

Thank you for the observation. We agree that this point needs clarification. Please kindly see the response to reviewer #2.

Method

Please define BPD and provide references.

Thank you for your observation. Please kindly see the added text in the Introduction section (page 3, lines 48-51) and Materials and Methods section (page 7, lines 145, 146).

Discussion

Overall, there is not enough discussion.

Please describe the mechanism by which cytokines including IL-6 are involved in BPD. Are there any previous reports on the association between nasal IL-6 and BPD? If there are no previous reports examining the association between nasal IL-6 and BPD, it would further enhance the value of this study.

Thank you for the useful suggestion. We have reviewed the literature and added some new information in the Discussion section, page 12 (lines 241-248 ). As you very well said, to our knowledge, NA-IL6 has not been studied yet in the context of BPD. We added this information in page 12, lines 230-231.

Please describe the safety of nasal discharge collection in very low birth weight infants. Because the nasal cavity is narrow, it would be helpful to describe whether there were any complications when inserting the catheter into the nasopharynx (e.g., nasal cavity damage, nosebleeds, desaturation, etc.). Also, please mention the product name of the catheter used.

The procedure and catheter used are the same as for the routine nasopharyngeal suctioning performed in the Neonatal Unit (5 Fr soft catheter made of silicone). Although the nasal cavity in preterm infants is narrow, so is the catheter and we did not observe any complications. We have clarified this information in the text (Material and Methods section, page 6, lines 123-124 and Results section, page 8 lines 172-173)

Reviewer #4

This is a clinical study conducted to provide lung ultrasound and nasopharyngeal interleukin-6 would be predicting marker of the development of BPD.

The topic is important and the manuscript is well written, but there are some concerns.

Major points

1. Concerns about very limited sample size. Further, the author should perform power analysis.

We planned this study as a pilot study or as a proof of concept, to explore if preterm infants expressed different IL6 values in the nasopharynx according to the diagnosis of BPD. The sample size was also determined by budget constraints. However, despite its small sample size, we have been able to find differences between patients with and without BPD according to the IL6 levels in nasopharyngeal secretions.

We have discussed the sample size limitations in the manuscript. Please kindly see the response to reviewer #2.

2. Chorioamnionitis is known to be associated with BPD. The author has to clarify the relationship between chorioamnionitis and NA IL-6 in this study. This leads whether NA IL-6 elevation is due to prenatal or postnatal factors.

Thank you for the comment. As reviewer #4 mentions, chorioamnionitis is known to be associated with BPD. It is an inflammation in the fetal membranes or placenta and when it occurs, fetal lungs are exposed to inflammatory cytokines and there is a higher risk of developing BPD. Amniotic fluid IL-6 has been proven to be a marker of intra-amniotic infection [*], but there is no literature about the relationship between amniotic fluid IL6 and NA IL-6 and BPD. We have not investigated this in our study either and, as we already mentioned in the manuscript, this is one of our limitations.

As you have identified, this is a very important question that we will try to address in future studies.

[*] Choi CW. Chorioamnionitis: Is a major player in the development of bronchopulmonary dysplasia? Korean J Pediatr. 2017 Jul;60(7):203-207. doi: 10.3345/kjp.2017.60.7.203

3. Multivariate analysis should be needed to clarify whether LUS and IL-6 associate with BPD.

Unfortunately, we don´t have enough sample size to do a multivariate analysis. As we answered to Reviewer #2, our aim in the future is to conduct a larger study and investigate the relationship between BPD and other covariates.

4. No description about Figure 3 was found in discussion.

Thank you for the observation. We have mentioned Figure 3 where appropriate in the text (Discussion section, page 11, line 218).

Minor points

1. Figure 2: There are no notations of the median or mean.

Please kindly see the added information in Figure 2.

2. Figure 3: How many samples did the author investigated NA-IL6? There are almost 35 dots in this Figure.

As we specified in Figure 1, we obtained 42 samples of IL6 which were investigated. We have checked our statistical analysis and the graphs we performed using the SPSS program and the sample size is 42. We think that some dots may be superimposed when represented in this graphic making it difficult to identify them.

3. SGA was associated with risk of BPD in this study?

We are a reference center for early and severe IUGR in Barcelona, this fact explains that of the 42 studied preterm infants, 14 of them were IUGR, representing 33% of the sample. All of them were born with a weight under the 10th percentile.

According to the literature, those patients are at higher risk of developing BPD, as we also found in our study (8/14, 57%), but we did not find statistically significant differences due to the small numbers. We did not find significant differences in IL6 values according to the IUGR diagnosis (p=0.345). These patients are born prematurely mainly due to their growth condition and not due to an intraamniotic infection or chorioamnionitis. The fact that the patients evolving to BDP (including IUGR patients) exhibit higher levels of IL6, may indicate that the inflammation found in nasopharyngeal aspirates is a reflex of their postnatal respiratory status and not related to perinatal factors.

---

## [Decision Letter · Decision Letter 1]

7 Feb 2025

Accuracy of point-of-care nasopharyngeal Interleukin 6 and lung ultrasound in predicting the development of bronchopulmonary dysplasia in preterm infants born before 30 weeks of gestation.

PONE-D-24-31846R1

Dear Dr. Aldecoa-Bilbao,

We’re pleased to inform you that your manuscript has been judged scientifically suitable for publication and will be formally accepted for publication once it meets all outstanding technical requirements.

Kind regards,

Kazumichi Fujioka

Academic Editor

PLOS ONE

Additional Editor Comments (optional):

Reviewers' comments:

Reviewer's Responses to Questions

**Comments to the Author**

1. If the authors have adequately addressed your comments raised in a previous round of review and you feel that this manuscript is now acceptable for publication, you may indicate that here to bypass the “Comments to the Author” section, enter your conflict of interest statement in the “Confidential to Editor” section, and submit your "Accept" recommendation.

Reviewer #1: (No Response)

Reviewer #3: All comments have been addressed

2. Is the manuscript technically sound, and do the data support the conclusions?

Reviewer #1: Yes

Reviewer #3: Yes

3. Has the statistical analysis been performed appropriately and rigorously? 

Reviewer #1: Yes

Reviewer #3: Yes

4. Have the authors made all data underlying the findings in their manuscript fully available?

Reviewer #1: Yes

Reviewer #3: Yes

5. Is the manuscript presented in an intelligible fashion and written in standard English?

Reviewer #1: Yes

Reviewer #3: Yes

6. Review Comments to the Author

Reviewer #1: Authors did a good job in revising the manuscript. .

Reviewer #3: I felt that the authors had properly responded to my initial peer review and revised the main text.

They had also responded to the comments of the other reviewers, and the paper had improved as a result.

7. PLOS authors have the option to publish the peer review history of their article (what does this mean? ). If published, this will include your full peer review and any attached files.

**Do you want your identity to be public for this peer review?** For information about this choice, including consent withdrawal, please see our Privacy Policy .

Reviewer #1: No

Reviewer #3: No

---

## [Editor Report · Acceptance letter]

PONE-D-24-31846R1

PLOS ONE

Dear Dr. Aldecoa-Bilbao,

I'm pleased to inform you that your manuscript has been deemed suitable for publication in PLOS ONE. Congratulations! Your manuscript is now being handed over to our production team.

Kind regards,

on behalf of

Dr. Kazumichi Fujioka

Academic Editor

PLOS ONE